# Responses of Tomato Photosystem II Photochemistry to Pegylated Zinc-Doped Ferrite Nanoparticles

**DOI:** 10.3390/nano15040288

**Published:** 2025-02-13

**Authors:** Ilektra Sperdouli, Kleoniki Giannousi, Julietta Moustaka, Orestis Antonoglou, Catherine Dendrinou-Samara, Michael Moustakas

**Affiliations:** 1Institute of Plant Breeding and Genetic Resources, Hellenic Agricultural Organization-Dimitra, 57001 Thessaloniki, Greece; 2Laboratory of Inorganic Chemistry, Department of Chemistry, Aristotle University of Thessaloniki, 54124 Thessaloniki, Greece; klegia@chem.auth.gr (K.G.); orestis1911@gmail.com (O.A.); samkat@chem.auth.gr (C.D.-S.); 3Department of Botany, School of Biology, Aristotle University of Thessaloniki, 54124 Thessaloniki, Greece

**Keywords:** magnetic nanoparticles, foliar spray, photosynthesis, hormesis, chlorophyll fluorescence, excess excitation energy, non-photochemical quenching, electron transport, excitation pressure

## Abstract

Various metal-based nanomaterials have been the focus of research regarding their use in controlling pests and diseases and in improving crop yield and quality. In this study, we synthesized via a solvothermal procedure pegylated zinc-doped ferrite (ZnFer) NPs and characterized their physicochemical properties by X-ray diffraction (XRD), vibrating sample magnetometry (VSM), thermogravimetric analysis (TGA), FT-IR and UV–Vis spectroscopies, as well as transmission electron microscopy (TEM). Subsequently, their impact on tomato photosynthetic efficiency was evaluated by using chlorophyll *a* fluorescence imaging analysis to estimate the light energy use efficiency of photosystem II (PSII), 30, 60, and 180 min after foliar spray of tomato plants with distilled water (control plants) or 15 mg L^−1^ and 30 mg L^−1^ ZnFer NPs. The PSII responses of tomato leaves to foliar spray with ZnFer NPs showed time- and dose-dependent biphasic hormetic responses, characterized by a short-time inhibitory effect by the low dose and stimulatory effect by the high dose, while at a longer exposure period, the reverse phenomenon was recorded by the low and high doses. An inhibitory effect on PSII function was observed after more than ~120 min exposure to both ZnFer NPs concentrations, implying a negative effect on PSII photochemistry. We may conclude that the synthesized ZnFer NPs, despite their ability to induce hormesis of PSII photochemistry, have a negative impact on photosynthetic function.

## 1. Introduction

Nanoparticles (NPs) have been shown to improve plant performance, suppress plant disease, and protect crop plants from environmental stresses [1,2]. However, some metal-based NPs have been described to decrease plant growth and performance, causing toxicity effects on crops [3]. Metal-based nanoparticles (NPs) are emerging as promising tools in sustainable agriculture and, nowadays, are being used as nanofertilizers, nanopesticides, growth stimulators, or simply as nanocarriers, since they are more effective and pose a lower risk of environmental contamination than conventional agrochemicals. Photosynthesis is strongly related to plant productivity and yield [4,5]. Despite the fact that photosynthesis is strongly related to plant productivity and yield, only a limited number of studies have focused on the use of NPs to improve photosynthetic activity [4,5].

Among the several metal-based NPs, the zinc oxide ones have been shown to have the potential to improve the photosynthetic efficacy of tomato plants, depending though on their shape; ZnO nanorods were not as effective as irregularly shaped ZnO NPs in enhancing PSII function [6,7]. Moreover, iron-based NPs are gaining increasing interest due to their redox chemistry and the specific property of zero-valent iron (Fe-0) NPs to promote stomatal opening, as observed in *Arabidopsis thaliana* [8]. Iron-based NPs are commonly synthesized in the form of ferrites M(Fe_x_O_y_), where M stands for metal [9]. Ferrites are magnetic nanomaterials, which apart from their rich physical, structural, and electrical properties, have the ability to be traced in an organism and directed externally by magnets [10]. Currently, magnetic nanoparticles are drawing attention owing to their wide-ranging applications across various fields, spanning from medicine and electronics to cosmetics and environmental remediation [11,12,13,14]. Meanwhile, based on their composition, M(Fe_x_O_y_) have been explored as biostimulants and photocatalysts in agricultural settings [5,11]. Specifically, zinc ferrites (ZnFe_2_O_4_) have semiconducting properties and can be considered photoreactive materials, including photocatalysis and photochemical reactions. Zinc ferrite NPs have been explored for their photocatalytic activity under UV or visible light [15,16]. In photocatalysis, those NPs can absorb light and generate electron–hole pairs that are capable of driving photochemical reactions, such as the degradation of organic pollutants or the production of hydrogen from water splitting. Meanwhile, the surface of ZnFe_2_O_4_ can be activated under light, which leads to the formation of reactive oxygen species (ROS), such as hydroxyl radicals (·OH), which can interact with organic contaminants, making them useful in environmental remediation and wastewater treatment.

The rising demands for sustainable agricultural practices to boost crop growth, enhance nutrient utilization, and increase resilience to abiotic and biotic stresses highlight nanofertilizers as valuable tools for regenerative agriculture [17]. Foliar application of nanofertilizers can not only enhance nutrient use efficiency [18] but also stimulate photosynthesis [6] and relieve the unfavorable effects of abiotic stresses [19]. Photosynthetic function is usually evaluated using the non-destructive method of chlorophyll *a* fluorescence [20,21,22,23,24]. The fluorescence of chlorophyll *a*, which can be easily evaluated, is a small fraction of one of the de-excitation pathways of absorbed light energy by light-harvesting complexes that is not used for photochemistry or dissipated through other pathways [25,26]. The absorbed light energy that is not utilized for photochemistry or dissipated as heat by a photoprotective mechanism of non-photochemical quenching (NPQ) can lead to reactive oxygen species (ROS) formation, such as superoxide anions (O_2_^•−^), hydroxyl radicals (OH^−^), singlet oxygen (^1^O_2_), and hydrogen peroxide (H_2_O_2_) [27,28,29]. The increased ROS generation can lead to oxidative damage but they also act as signaling molecules, playing functions in plant defense and plant development [29,30,31,32,33,34,35].

Plants respond to any disturbance of their homeostasis triggered by an abiotic or biotic stress factor, displaying a compensatory response that results in a hormetic response [36,37,38,39]. Accordingly, the term hormesis refers to an “overcompensation” response when there is a disruption in homeostasis, and it has been shown to occur in a number of organisms independent of the type of stress or the metabolic process considered [17,36,37,39,40,41,42,43,44,45,46,47,48,49]. Hormesis typically follows an inverted U-shaped response curve, showing stimulation at low doses or brief exposures and inhibition at high doses or prolonged exposures [46,50,51]. Conversely, a U-shaped biphasic response curve has also been observed, where low doses or short exposures result in inhibition, while higher doses or longer exposures lead to stimulation [37,39,40,52,53]. The hormetic response of photosynthetic function has been reported to involve activation of the NPQ mechanism [37,40,53,54,55].

Zinc (Zn), an essential micronutrient for plants, is required for normal plant growth and development [56,57,58]. It is involved in retaining the membrane structure of various cell organelles [59,60,61,62,63,64] and also participates as a structural component of proteins involved in the metabolism of ROS, in the process of translation and transcription, and in photosynthesis [58,65,66]. Zinc cations and Zn complexes have antioxidant and antimicrobial activities [67]. Iron (Fe) has a fundamental role in the enzymes involved in the electron transport of the photosynthetic process [68,69].

Human Zn nutrition is based on plants and human Zn deficiency results in over 400,000 deaths yearly [66]. Agricultural practices through foliar spray to enhance Zn bioavailability in nutritive systems [70,71,72] have become gradually common in regions of arid countries [73]. Iron and Zn deficiencies, due to the nutrient-poor status of agricultural soils, persist as serious public health problems affecting more than 3 billion people worldwide [74]. Agricultural application of Zn, in the framework of hormesis, for improving crop yield and quality can be somewhat outside laboratory interests especially when applied in addition to other benefits (e.g., control of pests and diseases) [75]. Tomato, as an extensively cultivated high-value vegetable, unavoidably faces various types of pests and diseases that not only severely impact the quality of tomatoes but also significantly decrease harvests, leading to considerable commercial losses [38,76].

Herein, in an attempt to combine two bio-metals with proved crop enhancing and protecting properties, zinc ferrite nanoparticles were solvothermally prepared in the presence of polyethylene glycol 8000 (PEG 8000) and their impact on tomato photosystem II photochemistry was evaluated. Zinc ferrite has a moderate band gap, typically around 1.9–2.2 eV, which makes it responsive to both UV and visible light. This property allows it to participate in photoreactive processes, especially when used in photocatalytic systems [16]. Meanwhile, PEG can influence photochemistry through its effects as a stabilizer or structural component in various systems. It can modulate reaction rates, enhance stability, protect sensitive molecules, and alter the properties of photoreactive materials [77]. Its impact on photochemistry is highly dependent on the specific context of the system and the nature of the photochemical processes involved [78]. We applied a synthetic procedure that allowed the isolation of coated/pegylated ferrites with a different zinc doping from the typical numerical formula (ZnFe_2_O_4_) to further test the optical, photocatalytic, and magnetic properties of the nanomaterial. The physicochemical characteristics of the resulting hybrid ensembles, which are very important for their biological evaluation, were identified. Since previous studies suggested the safe use of ZnFe_2_O_4_ NPs in nano-agricultural applications [9], we evaluated the consequences of our synthesized zinc ferrite NPs on tomato photosynthetic function 30 min, 60 min, and 180 min after foliar spray. Photosynthetic function measurements were conducted with two light intensities, 300 μmol photons m^−2^ s^−1^, corresponding to the growth light intensity (GLI) of tomato plants, and 1000 μmol photons m^−2^ s^−1^, representing a high light intensity (HLI).

## 2. Materials and Methods

### 2.1. Synthesis of Zinc-Doped Ferrite Nanoparticles

First, 1.8 mmol Fe(acac)_3_ and 0.9 mmol Zn(acac)_2_ were mixed in 10 mL of PEG 8000. The solution was transferred to a Teflon-lined autoclave and the reaction was carried out under autogenous pressure at 200 °C for 24 h (heating rate 4 °C/min). Subsequently, the mixture was centrifuged at 5000 rpm for 20 min, followed by rinsing with ethyl alcohol four times to remove any untreated precursors.

### 2.2. Characterization of Zinc-Doped Ferrite Nanoparticles

The crystal structure of the magnetic nanoparticles (MNPs) was investigated through X-ray diffraction (XRD) performed on a Philips PW 1820 diffractometer at a scanning rate of 0.050/3 s, in the 2θ range from 20° to 70°, with Cu Kα radiation (λ = 1.5406 nm). Magnetic measurements were performed using a vibrating sample magnetometer (VSM) (P.A.R. 155). Thermogravimetric analysis (TGA) was performed using a SETA-RAM SetSys-1200 (KEP Technologies, Caluire-et-Cuire, France) and carried out in the range from room temperature to 800 °C at a heating rate of 10 °C min^−1^ under an N_2_ atmosphere. Primary particle size and morphology were determined through conventional transmission electron microscopy (TEM) images obtained with a JEOL JEM 1200–EX (JEOL Ltd., Tokyo, Japan) microscope operating at 120 kV. For the TEM observations, suspensions deposited onto carbon-coated copper grids were used.

### 2.3. Plant Material and Growth Conditions

In our experiments, we used 20 cm in height tomato (*Solanum lycopersicum* L. cv. Meteor) plants grown in a greenhouse with a 14 h photoperiod, a photosynthetic photon flux density (PPFD) of 300 ± 10 μmol photons m^−2^ s^−1^, a day/night temperature 25 ± 1/21 ± 1 °C, and a relative humidity of 55 ± 5/65 ± 5% day/night.

### 2.4. Plant Treatment with Zinc-Doped Ferrite Nanoparticles

Two concentrations of ZnFer NPs (15 mg L^−1^ and 30 mg L^−1^) were used for foliar application in tomato plants. Control plants were foliar sprayed with distilled water. Each tomato plant was foliar sprayed with 15 mL of distilled water or ZnFer NPs. Three plants were used in each treatment with two independent replications (*n* = 6).

### 2.5. Chlorophyll Fluorescence Measurements

To estimate PSII function in young tomato leaflets, we used the Imaging-PAM Fluorometer M-Series (Heinz Walz GmbH, Effeltrich, Germany), as previously described in detail [79]. Chlorophyll fluorescence measurements were conducted 30 min, 60 min, and 180 min after exposure to distilled water (control plants) or 15 mg L^−1^ and 30 mg L^−1^ ZnFer NPs. All leaves were dark-adapted for 20 min before measuring the minimum (F*o*) and the maximum (F*m*) chlorophyll *a* fluorescence in the dark. The maximum chlorophyll *a* fluorescence in the light (F*m*′) was obtained with saturating pulses (SPs) every 20 s for 5 min after application of the actinic light (AL), while the minimum chlorophyll *a* fluorescence in the light (F*o*′) was computed as follows: F*o*′ = F*o*/(F*v*/F*m* + F*o*/F*m*′) [80]. The actinic light (AL) for the measurements was 300 μmol photons m^−2^ s^−1^, corresponding to the growth light intensity (GLI), or 1000 μmol photons m^−2^ s^−1^, representing a high light intensity (HLI). Steady-state photosynthesis (F*s*) was measured after 5 min of illumination time with the AL. The chlorophyll fluorescence parameters, described in detail in Appendix A, were estimated using Imaging Win V2.41a software (Heinz Walz GmbH, Effeltrich, Germany).

### 2.6. Statistical Analysis

All statistical analysis was conducted in R software (version 4.3.1, R Core Team, 2023). The normality of residuals was tested using the Shapiro–Wilk test (package ‘stats’) and homogeneity of variances was checked with Levene’s test (package ‘car’). A two-way ANOVA was performed for each parameter with concentration and time as factors, followed by post hoc analysis with Tukey’s honest significant difference test. Values were considered significantly different at *p* < 0.05.

## 3. Results

### 3.1. Synthesis and Characterization of Zinc-Doped Ferrite Nanoparticles

The composition, crystal structure, and crystallite size of the nanoparticles were identified via X-ray diffraction (XRD) analysis (Figure 1a). The XRD pattern verified the composition of zinc ferrite nanoparticles (#22-1012) without any impurities, while the exact stoichiometry was found to be Zn_0.65_Fe_0.35_Fe_2_O_4_ based on the Rietveld refinement analysis of the XRD pattern (Appendix A).

PEG was crystallized as indicated by the peaks presented in the diffractogram (#49-2095). The lattice parameter α was calculated to be 0.84 nm, and the crystallite size was found to be equal to 13 nm upon applying the Scherrer equation. The morphology of the sample was investigated by transmission electron microscopy (TEM) analysis (Figure 1b). PEG resulted in the formation of groups of almost spherical nanoparticles, while the Feret diameter was found to be approximately 12 nm (Figure 1b inset), close to the crystallite size. The organic coating of the ferrite was initially analyzed by FT-IR spectroscopy and the spectrum normalized in the region of 1800 to 400 cm^−1^ is presented in Figure 1c. The polymeric character of PEG (1150–1000 cm^−1^) was preserved and dominated over the oxidized forms of PEG (1700–1500 cm^−1^). The bands corresponding to CH_2_ and CH_3_ groups seemed to be of similar intensity, and thus it was deduced that limited fragmentation took place. The band due to metal–oxygen bonds was significantly attenuated in comparison to the bands of the organic coating, verifying that a high ratio of PEG was present in the sample. This observation was also confirmed by the thermogravimetric analysis (TGA) curve, where the total amount of PEG was found to be equal to 70% *w*/*w* (Figure 1d). The decomposition took place in two stages, indicating the presence of a double layer. The hysteresis loop, as measured by VSM and upon correction based on the percentage of surfactant, is presented in Figure 1e. The saturation magnetization (M_s_) was found to be equal to 104 emu/g. The optical properties of the zinc-doped ferrite NPs were investigated via UV–Vis measurements (Appendix A), and by employing a Tauc plot, the band gap was ascertained to be 2.35 eV (Appendix A), which was slightly larger than that of the bulk counterpart (1.9–2.2 eV).

### 3.2. Maximum Efficiency of Photosystem II and Efficiency of the Oxygen-Evolving Complex in Tomato Leaflets Sprayed with Zinc-Doped Ferrite Nanoparticles

Since there were no statistically significant differences after 30 min, 60 min, and 180 min between any of the control chlorophyll fluorescence parameters, one control average value is presented for simplicity.

The maximum efficiency of PSII photochemistry (F*v*/F*m*), measured in dark-adapted tomato leaves, 30 min after exposure to 15 mg L^−1^ ZnFer NPs, decreased significantly compared to exposure to distilled water (control plants), while F*v*/F*m* after exposure to 30 mg L^−1^ zinc ferrite NPs was equal to the level of control plants (Figure 2a). Exposure of tomato plants for 60 and 180 min to both ZnFer NPs concentrations resulted in decreased F*v*/F*m* values compared to control plants (Figure 2a). Tomato plants after 180 min exposure to 15 mg L^−1^ ZnFer NPs retained higher F*v*/F*m* values compared to exposure to 30 mg L^−1^ ZnFer NPs (Figure 2a).

The efficiency of the oxygen-evolving complex (F*v*/F*o*) was affected by the ZnFer NPs similarly to F*v*/F*m*. However, 30 and 180 min after foliar spray with 15 mg L^−1^ ZnFer NPs, the F*v*/F*o* values were equal to those of control plants (Figure 2b). By contrast, 180 min after foliar spray with 30 mg L^−1^ ZnFer NPs, the F*v*/F*o* values decreased significantly compared to the control plant values (Figure 2b).

### 3.3. Distribution of the Absorbed Light Energy in Photosystem II in Tomato Leaflets Sprayed with Zinc-Doped Ferrite Nanoparticles

The partitioning of the light energy at PSII to photochemistry (Φ_PSII_), heat dissipation lost (Φ_NPQ_), or misplaced through a nonregulatory pathway (Φ_NO_) [81] was measured 30 min, 60 min, and 180 min after exposure to distilled water (control plants) or 15 and 30 mg L^−1^ ZnFer NPs. Thirty minutes after exposure to 15 mg L^−1^ ZnFer NPs, the effective quantum yield of PSII photochemistry (Φ_PSII_), at the GLI and the HLI (Figure 3a,b), decreased significantly, while with 30 mg L^−1^ zinc ferrite NPs, Φ_PSII_ increased significantly, compared to control plants (Figure 3a,b). On the contrary, 60 min after exposure to 15 mg L^−1^ ZnFer NPs, Φ_PSII_ increased significantly at the GLI compared to control plants and was equal to controls at the HLI, while with 30 mg L^−1^ ZnFer NPs, Φ_PSII_ returned to control values at both light intensities (Figure 3a,b). Further exposure time (180 min), at both light intensities, resulted in a decreased Φ_PSII_ with both 15 mg L^−1^ and 30 mg L^−1^ ZnFer NPs compared to control plants (Figure 3a,b). However, at both light intensities, tomato plants after 180 min exposure to 15 mg L^−1^ ZnFer NPs retained higher Φ_PSII_ values compared to exposure to 30 mg L^−1^ ZnFer NPs, but the values were still lower than the control values (Figure 3a,b).

Τhe quantum yield of regulated non-photochemical energy loss in PSII (Φ_NPQ_) increased after exposure to 15 mg L^−1^ or 30 mg L^−1^ ZnFer NPs to the same level at both concentrations and at both light intensities (Figure 4a,b), as compared to control plants. At both concentrations and both light intensities, the higher increase in Φ_NPQ_ was observed after 180 min exposure to ZnFer NPs (Figure 4a,b).

The quantum yield of non-regulated energy loss in PSII (Φ_NO_), after 30 min exposure to 15 mg L^−1^ ZnFer NPs, remained at the level of control plants at both light intensities, but with 30 mg L^−1^ ZnFer NPs, Φ_NO_ decreased significantly compared to control plants (Figure 4c,d). After 60 and 180 min exposure to both ZnFer NPs concentrations, Φ_NO_ at both light intensities decreased significantly compared to control plants (Figure 4c,d). After 180 min exposure of tomato plants to 15 mg L^−1^ ZnFer NPs, Φ_NO_ at both light intensities retained lower values compared to exposure to 30 mg L^−1^ ZnFer NPs (Figure 4c,d).

### 3.4. The Fraction of Open PSII Reaction Centers and the Electron Transport Efficiency in Tomato Leaflets Sprayed with Zinc-Doped Ferrite Nanoparticles

The fraction of open PSII rection centers (RCs) (q*p*), after 30 min exposure to 15 mg L^−1^ ZnFer NPs, at both light intensities, decreased significantly (Figure 5a,b), while with 30 mg L^−1^ zinc ferrite NPs, q*p* increased significantly, compared to control plants (Figure 5a,b). However, after 60 min exposure to 15 mg L^−1^ ZnFer NPs, q*p* at both light intensities was higher than exposure to 30 mg L^−1^ zinc ferrite NPs and in control plants (Figure 5a,b). After 180 min exposure to 15 mg L^−1^ ZnFer NPs, q*p* was higher at both light intensities compared to exposure to 30 mg L^−1^ zinc ferrite NPs and equal to the level of control plants (Figure 5a,b).

The electron transport rate (ETR), after 30 min exposure to 15 mg L^−1^ ZnFer NPs, at both light intensities, decreased significantly, while with 30 mg L^−1^ zinc ferrite NPs, ETR increased significantly at both light intensities, compared to control plants (Figure 6a,b).

On the contrary, 60 min after exposure to 15 mg L^−1^ ZnFer NPs, ETR increased significantly at the GLI compared to control plants and was equal to controls at the HLI, while with 30 mg L^−1^ ZnFer NPs, Φ_PSII_ returned to control values at both light intensities (Figure 6a,b). Further exposure time (180 min), at both light intensities, resulted in a decreased ETR at both concentrations compared to control plants (Figure 6a,b). However, at both light intensities, tomato plants after 180 min exposure to 15 mg L^−1^ ZnFer NPs retained higher ETR values compared to exposure to 30 mg L^−1^ ZnFer NPs, but the values were still lower than the control values (Figure 6a,b).

### 3.5. Impact of Zinc-Doped Ferrite Nanoparticles on the Efficiency of PSII Reaction Centers and the Non-Photochemical Quenching

The efficiency of the open PSII RCs (F*v*′/F*m*′) at the GLI, after exposure for 30 min, 60 min, and 180 min to 15 and 30 mg L^−1^ ZnFer NPs, decreased significantly compared to control plants (Figure 7a). The same trend for F*v*′/F*m*′ was observed at the HLI (Figure 7b), with the only exception being after 30 min exposure to 15 mg L^−1^ ZnFer NPs, in which F*v*′/F*m*′ did not differ significantly from control plants (Figure 7b).

The non-photochemical quenching (NPQ), at both light intensities, with both ZnFer NPs concentrations, and under all exposure treatments, increased significantly compared to control plants (Figure 8a,b), with the only exception being after 30 min exposure to 15 mg L^−1^ ZnFer NPs at both light intensities, in which NPQ did not differ significantly compared to control plants (Figure 8b).

### 3.6. Impact of Zinc-Doped Ferrite Nanoparticles on PSII Excitation Presure and on Excess Excitation Energy at PSII

The excitation pressure at PSII (1−qL) at the GLI, after 30 min exposure to 15 mg L^−1^ ZnFer NPs, remained equal to control plants, while at all other treatments it was decreased compared to control plants (Figure 9a). At the HLI, after 30 min exposure to 15 mg L^−1^ ZnFer NPs, 1−qL increased compared to control plants (Figure 9b), but at all other treatments it was decreased compared to control plants, except it remained equal to control plants in the case of 180 min exposure to 30 mg L^−1^ ZnFer NPs (Figure 9b).

The excess excitation energy at PSII (EXC), at both light intensities, after 30 min exposure to 15 mg L^−1^ ZnFer NPs, increased compared to control plants, while with 30 mg L^−1^ ZnFer NPs it decreased compared to control plants (Figure 10a,b). After 60 min exposure at GLI, EXC decreased with exposure to 15 mg L^−1^ ZnFer NPs or remained equal to control plants with exposure to 30 mg L^−1^ ZnFer NPs (Figure 10a). At HLI, EXC remained equal to control plants with both ZnFer NPs treatments (Figure 10b). After 180 min exposure to both ZnFer NPs concentrations, EXC at both light intensities increased compared to control plants (Figure 10a,b).

### 3.7. Hormetic Responses of Photosystem II in Tomato Leaflets Sprayed with Zinc-Doped Ferrite Nanoparticles

The response of PSII photochemistry in tomato leaves to foliar spray with 15 and 30 mg L^−1^ ZnFer NPs, evaluated by the quantum yield of PSII photochemistry (Φ_PSII_) at the GLI and at the HLI, showed a time-dependent biphasic hormetic response. These biphasic hormetic response curves, which were almost similar at the GLI (Figure 11a) and at the HLI (Figure 11b), were characterized by a short-time (30 min) inhibitory effect by the low dose (15 mg L^−1^ ZnFer NPs) and by a stimulatory effect by the high dose (30 mg L^−1^ ZnFer NPs). At a longer period (60 min), a stimulatory effect was observed by the low dose and an inhibitory effect was observed by the high dose. Further exposure time (180 min) resulted in an inhibitory effect of PSII function at both concentrations. The stimulatory effect by 15 mg L^−1^ ZnFer NPs at the HLI after 60 min exposure was to a lesser extent and a shorter time duration (Figure 11b) than the stimulatory effect observed at the GLI (Figure 11a).

The time-dependent response curves of the excitation pressure (1−qL) in tomato leaves to foliar spray with 15 and 30 mg L^−1^ ZnFer NPs are displayed in Figure 12a for the GLI and in Figure 12b for the HLI. The response curves, which were almost similar at the GLI (Figure 12a) and at the HLI (Figure 12b), were characterized up to 30 min exposure time by an increased excitation pressure with the low dose (15 mg L^−1^ ZnFer NPs) and by a reduced excitation pressure up to 180 min exposure time with the high dose (30 mg L^−1^ ZnFer NPs). Further exposure time (from 30 to 180 min) of tomato leaves at the low dose (15 mg L^−1^ ZnFer NPs) reduced the excitation pressure (Figure 12a,b).

The time-dependent biphasic response curve of the excess excitation energy at the GLI (Figure 13a) was characterized (compared to control plants) to up to ~45 min exposure time by an increased excess excitation energy with the low ZnFer NPs dose, while at longer than ~45 min exposure time, it was characterized by a decreased excess excitation energy up to ~130 min, and at longer exposure time, it was characterized by an increased excess excitation energy (Figure 13a). The response curve of the excess excitation energy to the high ZnFer NPs dose at the GLI was characterized up ~75 min exposure time by a decreased excess excitation energy, which increased at longer exposure time (Figure 13a). At the HLI, the response curve of the excess excitation energy with the low ZnFer NPs dose was characterized up ~45 min exposure time by an increased excess excitation energy that decreased later, compared to up to ~120 min exposure time for control plants, and increased thereafter (Figure 13b). The response curve of the excess excitation energy with the high ZnFer NPs dose at the HLI was characterized up ~45 min exposure time by a decreased excess excitation energy, compared to control plants, which increased thereafter (Figure 13b).

## 4. Discussion

Zinc doping in the ferrite system is expected to enhance the photocatalytic potential of nanoparticles primarily in terms of the absorption of UV light and less effectively concerning the utilization of visible light for photochemical or photocatalytic processes [82]. Indeed, the band gap of ZnFer NPs was found to be slightly larger than that of the bulk ZnFe_2_O_4_, attributed either to nanosize effect or to zinc deficiency. However, the ZnFer NPs studied here presented a hybrid ensemble, since PEG formed a robust shell on the surface of NPs (70% *w*/*w*). PEGylation, from the one side, allows for better transportation and absorption of the NPs through the plants’ membranes due to stealth characteristics, but from the other side, it acts as hindrance to their photocatalytic activity.

The ability of crops to maintain their photosynthetic efficiency under environmental changes or when attacked by antagonists is crucial for plants [24,38,83,84,85,86]. High photosynthetic rates in chloroplasts require efficient light energy use [87,88], and thus it is generally recognized that light energy use efficiency, which is influenced by both the leaf age and the light intensity [87], controls crop yields [17,89,90,91,92,93,94]. Increasing photosynthetic efficiency lies in optimizing the allocation of absorbed light energy [95] and remains an important challenge for plant scientists, principally identified by the global rising need for nutrition [96,97,98].

The absorbed light energy that is driven to photochemistry, also identified as the effective quantum yield of PSII photochemistry (Φ_PSII_), was enhanced after 30 min exposure to 30 mg L^−1^ ZnFer NPs, compared to controls (Figure 3a,b). This was due to the increased fraction of open PSII reaction centers (RCs) (Figure 5a,b). However, at the same exposure period, Φ_PSII_ decreased with exposure to 15 mg L^−1^ ZnFer NPs (Figure 3a,b) due to a reduced fraction of open PSII RCs (Figure 5a,b). The decline in Φ_PSII_ can be ascribed (i) to a lower fraction of open PSII RCs (q*p*) or (ii) to a reduced efficiency of the RCs (F*v*′/F*m*′) [99]. After 60 min exposure, the reverse phenomenon occurred, with increased Φ_PSII_ with exposure to 15 mg L^−1^ ZnFer NPs (Figure 3a,b) due to an increased fraction of open PSII RCs (Figure 5a,b), which was equal to control plants and Φ_PSII_ with exposure to 30 mg L^−1^ ZnFer NPs, despite the increased fraction, compared to controls, of open PSII RCs (Figure 5a,b), but having a reduced efficiency of open PSII RCs (F*v*′/F*m*′) (Figure 7a,b). The decreased Φ_PSII_ after 180 min exposure to both concentrations at both light intensities (Figure 3a,b) was overcompensated by the increased non-photochemical energy loss in PSII (Φ_NPQ_) (Figure 4a,b) and, as a result, the non-regulated energy loss in PSII (Φ_NO_) decreased with both concentrations and at both light intensities compared to controls (Figure 4c,d). The non-regulated energy loss in PSII (Φ_NO_) represents the creation of singlet oxygen (^1^O_2_) via the triplet state of chlorophyll (^3^chl*) [100,101,102,103].

The photoprotective mechanism of non-photochemical quenching (NPQ), by dissipating the excess light energy as heat, prevents ROS formation [25,79,104,105,106,107,108,109,110,111,112,113,114]. The increased NPQ after 180 min exposure period to both ZnFer NPs concentrations, at both light intensities (Figure 8a,b), decreased the ^1^O_2_ generation (Figure 4c,d), and the higher increase in NPQ with exposure to 15 mg L^−1^ ZnFer NPs, compared to 30 mg L^−1^ ZnFer NPs (Figure 8a,b), induced a higher decrease in ^1^O_2_ generation (Figure 4c,d). NPQ can be considered to be sufficient if it can keep the fraction of open PSII RCs (q*p*) to the same level as under control conditions, under any disruption of plant homeostasis [115,116]. In accordance with this, the higher increase in NPQ with exposure to 15 mg L^−1^ ZnFer NPs, compared to 30 mg L^−1^ ZnFer NPs, after 180 min exposure time (Figure 8a,b) seemed to be adequate to keep the same portion of open PSII RCs (q*p*) as in controls and resulted in a higher decrease in ^1^O_2_ generation (Figure 4c,d). Thus, the increased NPQ altered ROS homeostasis through decreased ^1^O_2_ formation. Nanoparticles can contribute positively to cellular functions and signaling by generating low levels of ROS [117]. ROS produced during the light reactions of photosynthesis play essential roles as regulatory molecules in signaling processes, activating the plant’s protective defense response to environmental stressors as a result of environmental changes and contributing to restoring the “oxidation–reduction” balance [34,118,119,120,121,122,123,124]. Oxidative damage under biotic or abiotic stress can be avoided by joined prevention mechanisms and ROS detoxification [125,126,127,128].

Despite an inhibitory effect on PSII function after 180 min exposure period to both ZnFer NPs concentrations (Figure 3a,b), the excitation pressure at PSII (1−qL) was reduced (Figure 9a,b) due to the decreased generation of singlet oxygen (^1^O_2_) (Figure 4c,d) triggered by NPQ. Thus, foliar spray with ZnFer NPs caused an enhancement in NPQ, compared to control plants, which resulted in reducing the excitation pressure at PSII (1−qL) (Figure 9a,b).

Photosynthetic water oxidation is carried out in the OEC, which is located on the electron donor side of PSII, and consists of the so-called Mn_4_CaO_5_ cluster [129,130,131]. The electrons and protons produced by water oxidation are used to transform the light energy to ATP and NADPH for the synthesis of organic biomolecules needed by organisms [131,132]. The decreased efficiency of PSII photochemistry (F*v*/F*m*) 180 min after exposure to 30 mg L^−1^ ZnFer NPs, (Figure 2a) was related to the decreased efficiency of the OEC (Figure 2b), as judged from the ratio F*v*/F*o* [87,133,134,135,136,137,138,139,140]. A decreased efficiency of the OEC results in a decline in the F*v*/F*m* ratio [54,141,142], which represents the degree of plant photoinhibition [143]. Photoinhibition occurs when the light energy captured by light-harvesting complexes is in excess of what can be consumed by photochemistry, or dissipated safely by NPQ, or when the primary cause is the excitation of Mn in the OEC by photons [144,145,146,147,148,149,150]. Photoinhibition has been linked to a malfunction of the OEC [151,152,153,154]. If the OEC does not properly reduce the primary electron acceptor, then it can cause damaging oxidations at PSII [154,155,156,157].

The response of PSII photochemistry of tomato leaves to foliar spray with 15 and 30 mg L^−1^ ZnFer NPs, evaluated by the quantum yield of PSII photochemistry (Φ_PSII_) at the GLI (Figure 11a) and at the HLI (Figure 11b), showed a time-dependent biphasic hormetic response. This biphasic hormetic response curve was characterized by a short-time (30 min) inhibitory effect by a low dose (15 mg L^−1^ ZnFer NPs) and by a stimulatory effect by a high dose (30 mg L^−1^ ZnFer NPs). At longer period (60 min), a stimulatory effect was observed by the low dose and an inhibitory effect was observed by the high dose. Further exposure time resulted in an inhibitory effect of PSII function at both concentrations. Additionally, ZnFer nanoparticles showed an enhanced magnetization value of 104 emu/g compared to PEGylated zinc ferrite nanoparticles (M_s_ = 50 emu/g) reported in another study for MRI applications [158]. Their strong magnetic properties may influence the generation of ROS, which can either act as signaling molecules or cause oxidative stress. Excess ROS can damage PSII by oxidizing proteins and pigments involved in light harvesting, leading to photoinhibition [35]. The hormesis response of photosynthetic function has been reported to be triggered by the NPQ mechanism [37,42,45,79]. Hormesis appears to depend on the type of nanomaterial, with the same concentration inducing different responses with different nanomaterials [159].

The decreased Φ_PSII_ 180 min after foliar spray of tomato plants with ZnFer NPs (Figure 3a,b) resulted in a decreased ETR (Figure 6a,b), causing an increased excess excitation energy (EXC) (Figure 10a,b). Factors restricting ETR in plants result in the accumulation of excess excitation energy (EXC) at PSII [160,161,162].

## 5. Conclusions

The response of PSII photochemistry of tomato leaves to ZnFer NPs foliar spray showed a time-dependent biphasic hormetic response, characterized by a short-time inhibitory effect by the low dose and stimulatory effect by the high dose, while at a longer exposure period, a stimulatory effect was observed by the low dose and an inhibitory effect was observed by the high dose. An inhibitory effect on PSII function was observed after more than ~120 min exposure period to both ZnFer NPs concentrations. We may conclude that the synthesized ZnFer NPs, despite their ability to induce hormesis of PSII photochemistry, cannot be used as photosynthetic biostimulants. These findings align with previous results that showed both beneficial and detrimental effects of various metal oxide NPs (including zinc ferrites) on tomato fruit quality and suggested the need for caution in their applications in crops [4]. However, the results contribute valuable insights into understanding the biphasic response and the time-dependent thresholds at which NPs transition from being stimulatory to inhibitory, providing critical knowledge for future research on nanomaterials with improved functionality and reduced adverse effects.

## Figures and Tables

**Figure 1 nanomaterials-15-00288-f001:**
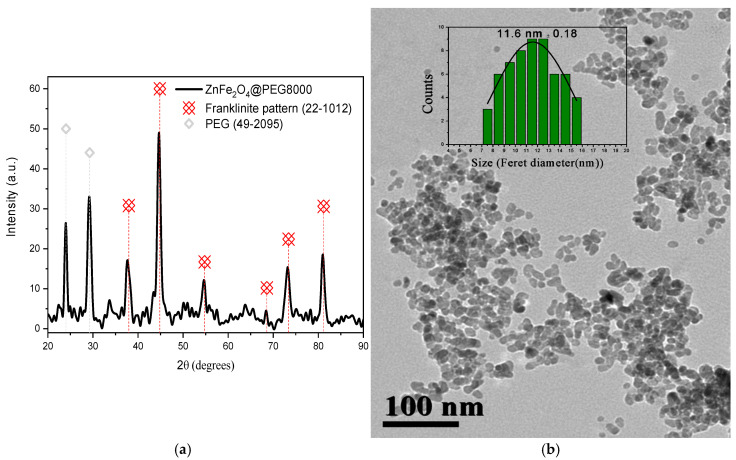
X-ray diffraction (XRD) pattern (**a**), TEM image with inset of a size distribution histogram and Gaussian fitting curve (**b**), FT-IR spectrum (**c**), thermogravimetric analysis (TGA) curve (**d**), and hysteresis loop by vibrating sample measurement (VSM) (**e**) of ZnFer NPs.

**Figure 2 nanomaterials-15-00288-f002:**
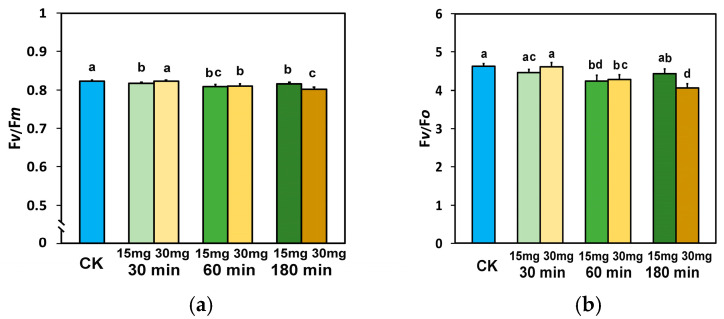
The maximum efficiency of PSII photochemistry (F*v*/F*m*) (**a**) and the efficiency of the oxygen-evolving complex (F*v*/F*o*) (**b**) in dark-adapted tomato leaves 30 min, 60 min, and 180 min after exposure to 15 mg L^−1^ and 30 mg L^−1^ ZnFer NPs, compared to control plants (CK). Bars represent standard deviations (SD). Different lower-case letters denote significant differences at *p* < 0.05 (*n* = 6).

**Figure 3 nanomaterials-15-00288-f003:**
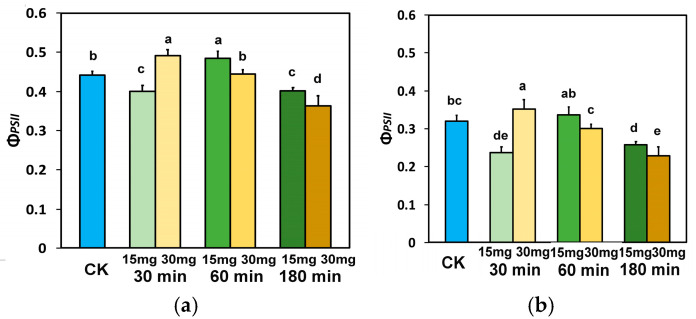
Light energy distribution at PSII. The effective quantum yield of PSII photochemistry (Φ_PSII_) measured at the growth light intensity (GLI) (**a**) and the effective quantum yield of PSII photochemistry (Φ_PSII_) measured at the high light intensity (HLI) (**b**) in tomato leaves, 30 min, 60 min, and 180 min after exposure to 15 mg L^−1^ and 30 mg L^−1^ ZnFer NPs, compared to control plants (CK). Bars represent standard deviations (SD). Different lower-case letters denote significant differences at *p* < 0.05 (*n* = 6).

**Figure 4 nanomaterials-15-00288-f004:**
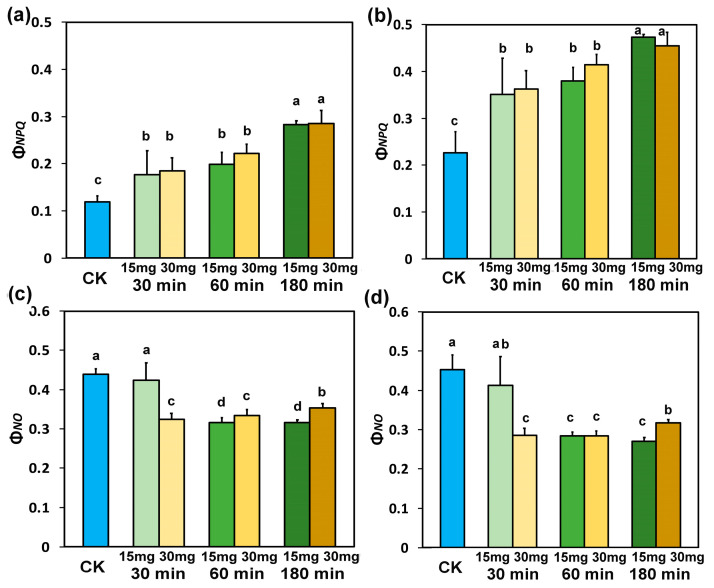
Light energy distribution at PSII. The quantum yield of regulated non-photochemical energy loss in PSII (Φ_NPQ_) measured at the GLI (**a**), the quantum yield of regulated non-photochemical energy loss in PSII (Φ_NPQ_) measured at the HLI (**b**), the quantum yield of non-regulated energy loss in PSII (Φ_NO_) measured at the GLI (**c**), and the quantum yield of non-regulated energy loss in PSII (Φ_NO_) measured at the HLI (**d**) in tomato leaves 30 min, 60 min, and 180 min after exposure to 15 mg L^−1^ and 30 mg L^−1^ ZnFer NPs, compared to control plants (CK). Bars represent standard deviations (SD). Different lower-case letters denote significant differences at *p* < 0.05 (*n* = 6).

**Figure 5 nanomaterials-15-00288-f005:**
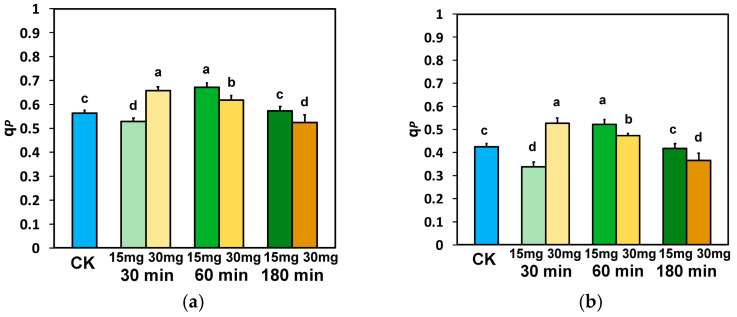
The fraction of open PSII rection centers (RCs) (q*p*) measured at the GLI (**a**) and at the HLI (**b**) in tomato leaves 30 min, 60 min, and 180 min after exposure to 15 mg L^−1^ and 30 mg L^−1^ ZnFer NPs, compared to control plants (CK). Bars represent standard deviations (SD). Different lower-case letters denote significant differences at *p* < 0.05 (*n* = 6).

**Figure 6 nanomaterials-15-00288-f006:**
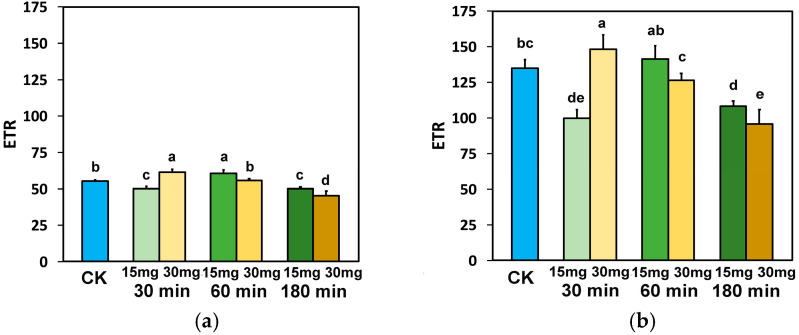
The electron transport rate (ETR) measured at the GLI (**a**) and at the HLI (**b**) in tomato leaves 30 min, 60 min, and 180 min after exposure to 15 mg L^−1^ and 30 mg L^−1^ ZnFer NPs, compared to control plants (CK). Bars represent standard deviations (SD). Different lower-case letters denote significant differences at *p* < 0.05 (*n* = 6).

**Figure 7 nanomaterials-15-00288-f007:**
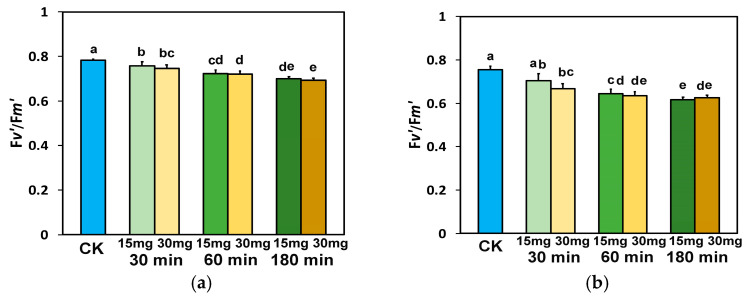
The efficiency of the open PSII RCs (F*v*′/F*m*′) measured at the GLI (**a**) and at the HLI (**b**) in tomato leaves 30 min, 60 min, and 180 min after exposure to 15 mg L^−1^ and 30 mg L^−1^ ZnFer NPs, compared to control plants (CK). Bars represent standard deviations (SD). Different lower-case letters denote significant differences at *p* < 0.05 (*n* = 6).

**Figure 8 nanomaterials-15-00288-f008:**
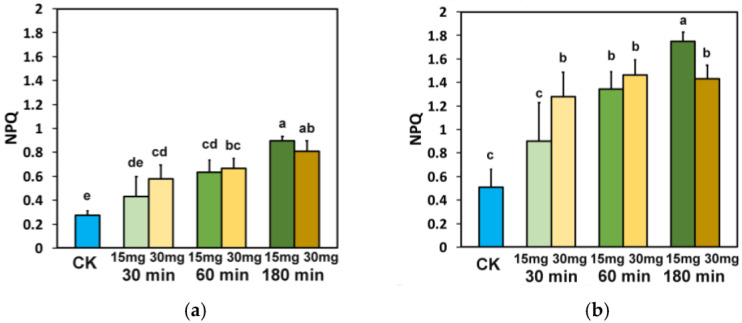
The non-photochemical quenching (NPQ) measured at the GLI (**a**) and at the HLI (**b**) in tomato leaves 30 min, 60 min, and 180 min after exposure to 15 mg L^−1^ and 30 mg L^−1^ ZnFer NPs, compared to control plants (CK). Bars represent standard deviations (SD). Different lower-case letters denote significant differences at *p* < 0.05 (*n* = 6).

**Figure 9 nanomaterials-15-00288-f009:**
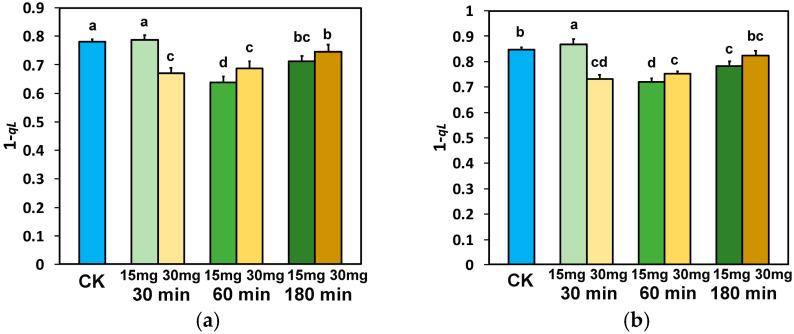
The excitation pressure at PSII (1−qL) measured at the GLI (**a**) and at the HLI (**b**) in tomato leaves 30 min, 60 min, and 180 min after exposure to 15 mg L^−1^ and 30 mg L^−1^ ZnFer NPs, compared to control plants (CK). Bars represent standard deviations (SD). Different lower-case letters denote significant differences at *p* < 0.05 (*n* = 6).

**Figure 10 nanomaterials-15-00288-f010:**
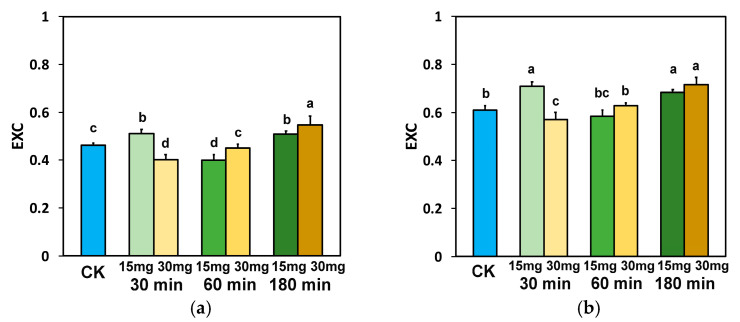
The excess excitation energy at PSII (EXC) measured at the GLI (**a**) and at the HLI (**b**) in tomato leaves 30 min, 60 min, and 180 min after exposure to 15 mg L^−1^ and 30 mg L^−1^ ZnFer NPs, compared to control plants (CK). Bars represent standard deviations (SD). Different lower-case letters denote significant differences at *p* < 0.05 (*n* = 6).

**Figure 11 nanomaterials-15-00288-f011:**
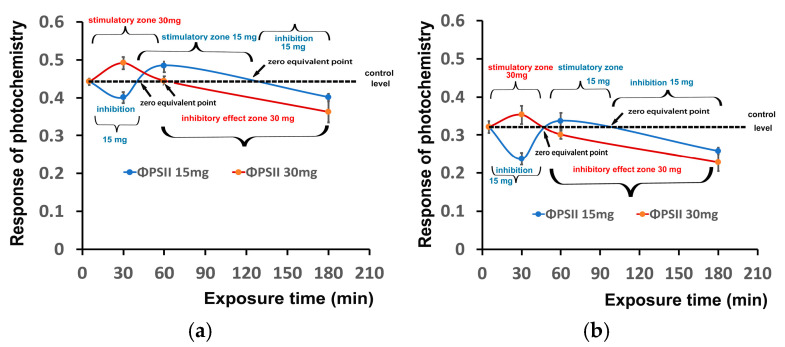
The time-dependent biphasic response curves of the quantum yield of PSII photochemistry (Φ_PSII_) measured at the GLI (**a**) and at the HLI (**b**), in tomato leaves 30 min, 60 min, and 180 min after exposure to 15 mg L^−1^ (blue line) and 30 mg L^−1^ (red line) ZnFer NPs, compared to the control level. Bars represent standard deviations (*n* = 6).

**Figure 12 nanomaterials-15-00288-f012:**
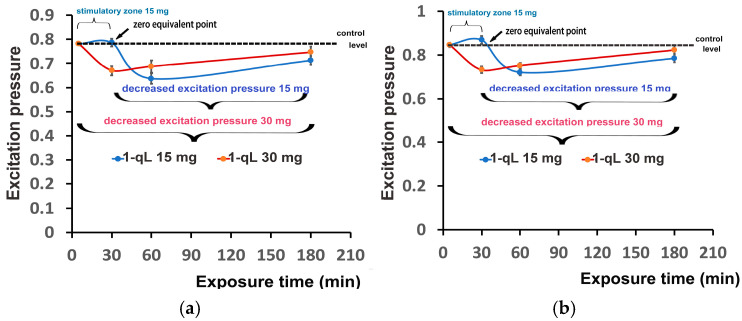
The time-dependent response curves of the excitation pressure at PSII (1−qL) measured at the GLI (**a**) and at the HLI (**b**) in tomato leaves 30 min, 60 min, and 180 min after exposure to 15 mg L^−1^ (blue line) and 30 mg L^−1^ (red line) ZnFer NPs, compared to the control level. Bars represent standard deviations (*n* = 6).

**Figure 13 nanomaterials-15-00288-f013:**
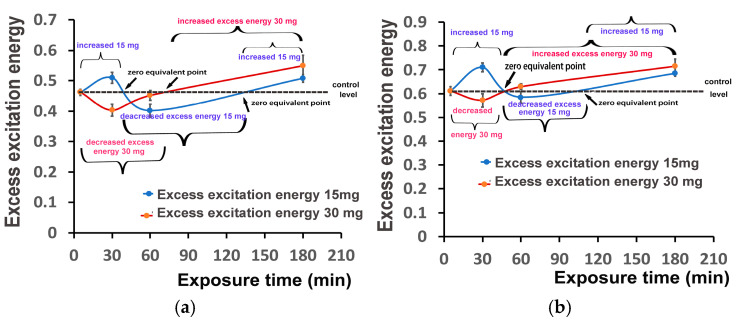
The time-dependent biphasic response curve of the excess excitation energy at PSII (EXC) measured at the GLI (**a**) and at the HLI (**b**) in tomato leaves 30 min, 60 min, and 180 min after exposure to 15 mg L^−1^ (blue line) and 30 mg L^−1^ (red line) ZnFer NPs, compared to the control level. Bars represent standard deviations (*n* = 6).

## Data Availability

All data supporting the findings of this study are available within the paper and within its Appendix A published online.

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
