# Peer review of "Responses of Tomato Photosystem II Photochemistry to Pegylated Zinc-Doped Ferrite Nanoparticles"

_nanomaterials, 2025, doi:10.3390/nano15040288_

Round 1
Reviewer 1 Report
Comments and Suggestions for Authors
The manuscript “Mechanistic Insights on Responses of Tomato Photosystem II Photochemistry to Pegylated Zinc-doped Ferrite Nanoparticles” by Sperdouli et al. explores the effect of pegylated zinc-doped ferrite nanoparticles on the photochemical activity of photosystem II in tomato plants. The topic is of high interest taking into account the implications, as well as the limitations, of the recent developments in nanoagronomy. The characterization of photosynthesis, a process of outmost importance for plants survival and adaptation, is properly selected in order to define the plants response to nanoparticles application. On the basis of the observed changes in photosystem II photochemistry and ability for photoprotection, the authors conclude that the selected nanoparticels exhibit inhibitory effect upon more than 120 min exposure and thus cannot be used as photosynthetic biostimulants.
The experimental data are scientifically sound and correctly interpreted. The critical discussion is well structured and the conclusions are also clearly defined. Thus, the paper is generally very well written and understandable.
However, I have a few minor comments that I believe would be beneficial for the paper quality:
1. As I understand, the authors idea is to use the pegylated zinc-doped ferrite nanoparticles in order to deliver Zn and/or Fe to the plant. Is that so? Do you have evidence that those elements are released from the nanoparticles? Could you provide more details on that matter?
2. From Fig. 1b it appears that the nanoparticles are mostly aggregates which would impair their incorporation into plant tissues. Can you comment on this effect? Generally, PEG is used to disaggregate nanoparticles but in this case it seems it does not do so.
3. Are the nanoparticles homogeneously distributed on the leave surface?
4. Could you provide some estimate of the amount of solution that you use for spraying per plant or leave?
5. Are there free PEG molecules in the nanoparticles dispersion that you use for spraying and if so what is their effect on photosynthesis.
6. Did you check the signal of nanoparticles themselves during PAM imaging experiment? Does it interfere with chl fluorescence?
Author Response
- As I understand, the authors idea is to use the pegylated zinc-doped ferrite nanoparticles in order to deliver Zn and/or Fe to the plant. Is that so? Do you have evidence that those elements are released from the nanoparticles? Could you provide more details on that matter?
Answer: The aim of the study has been given more clearly to the revised text as below. Actually, we didn’t proceed to such experiments due to our final disantavangeous effects on the photochemical activity of photosystem II in tomato plants.
Introduction Part
“Specifically, zinc ferrites (ZnFe₂O₄) have semiconducting properties and can be considered photoreactive materials, including photocatalysis and photochemical reactions. Zinc ferrites NPs have been explored for their photocatalytic activity under UV or visible light [15,16]. In photocatalysis, those NPs can absorb light and generate electron-hole pairs that are capable of driving photochemical reactions, such as the degradation of organic pollutants or the production of hydrogen from water splitting. Meanwhile, the surface of ZnFe₂O₄ can be activated under light, which leads to the formation of reactive oxygen species (ROS), such as hydroxyl radicals (·OH), that can interact with organic contaminants, making them useful in environmental remediation and wastewater treatment.
Herein, in an attempt, to combine two bio-metals with proved crop enhancing and protecting properties, zinc ferrite nanoparticles have been solvothermally prepared in the presence of polyethylene glycol 8000 (PEG 8000) and evaluated on tomato photosystem II photochemistry. Zinc ferrite has a moderate band gap, typically around 1.9–2.2 eV, which makes it responsive to both UV and visible light. This property allows it to participate in photoreactive processes, especially when used in photocatalytic systems. Meanwhile PEG can influence photochemistry through its effects as a stabilizer, or structural component in various systems. It can modulate reaction rates, enhance stability, protect sensitive molecules, and alter the properties of photoreactive materials [77]. Its impact on photochemistry is highly dependent on the specific context of the system and the nature of the photochemical processes involved [78]. We applied a synthetic procedure that allowed the isolation of coated/pegylated ferrites with a zinc doping different from the typical numerical formula (ZnFe2O4), to test further the optical, photocatalytic and magnetic properties of the nanomaterial.”
- Discussion
The zinc doping in the ferrite system is expected to enhance the photocatalytic potential of the nanoparticles primarily in terms of absorption of UV light and less effectively concerning utilizing visible light for photochemical or photocatalytic processes [82]. Indeed, the band gap of ZnFer NPs was found slightly larger than the bulk ZnFe2O4 attributed either to nanosize effect and/or to zinc deficiency. However, ZnFer NPs studied here present a hybrid ensemble, since PEG forms a robust shell on the surface of NPs (70% w/w). PEGylation from the one side allows the better transportation and absorption of the NPs through the plants’ membranes due to the stealth characteristics, but from the other side acts as hindrance in their photocatalytic activity.”
- Conclusions
“These findings meet previous results on different types of ferrites(ZnFe2O4 also included) where, in longer growth cycles, seems to be less of an effect from nanoparticles exposure on tomato plant physiological parameters [4]. Thus, both beneficial and detrimental effects in nutritional quality of tomato fruit were induced [4]. This highlights the complex relationship of the type of nanomaterial, concentration, plant species, and growth cycle duration that have to be taking into account in order to use further these nanomaterials for agrochemical purposes. However, the results contribute valuable insights into understanding the biphasic response and the thresholds at which NPs transition from being stimulatory to inhibitory provides critical knowledge for the design of future nanomaterials with improved functionality and reduced adverse effects.”
- From Fig. 1b it appears that the nanoparticles are mostly aggregates which would impair their incorporation into plant tissues. Can you comment on this effect? Generally, PEG is used to disaggregate nanoparticles but in this case it seems it does not do so.
Answer: Ferrite nanoparticles, particularly doped ferrites, often exhibit strong magnetic interactions. Even with PEGylation, these interactions can cause the particles to cluster partially, especially if the PEG coating does not fully mask the magnetic properties. (The role of PEG is given above at the revised text).
- Are the nanoparticles homogeneously distributed on the leave surface?
Answer: Indeed, the technique used to apply nanoparticles (e.g., spraying, dipping, brushing) affects their distribution. Spraying with aqueous well-dispersed droplets as in the present study is more likely to result in homogeneous coverage compared to other methods.
- Could you provide some estimate of the amount of solution that you use for spraying per plant or leave?
Answer: Thank you for your comment. We added in the manuscript “Each tomato plant was foliar sprayed with 15 ml distilled water or ZnFer NPs.
- Are there free PEG molecules in the nanoparticles dispersion that you use for spraying and if so what is their effect on photosynthesis.
Answer: There is no evidence of free PEG molecules. Based on the crystallization of PEG observed by XRD as well as by TGA measurements, where a well-formed layer of PEG ~ 57% is recorded and uniformly decomposed, we have no reason to believe that there is a significant portion of free PEG molecules.
- Did you check the signal of nanoparticles themselves during PAM imaging experiment? Does it interfere with chl fluorescence?
Answer: Chlorophyll a fluorescence emission results from the absorbed light energy by chlorophyll molecules that it is not dissipated as heat or not used for photosynthetic reactions at PSII in plants. Although chlorophyll a fluorescence in plants is only 0.6–5%, it offers valuable information about the partitioning of the absorbed light energy at PSII. The absorbed light energy is allocated to PSII photochemistry (ΦPSII), regulated non-photochemical energy in PSII (ΦNPQ), and non-regulated energy loss in PSII (ΦNO).
Thus, nanoparticles themselves during PAM imaging experiments, since they do not possess any chlorophyll molecules, they do not have any signal. Therefore we evaluate the effect of nanoparticles on photosynthesis by measuring the plant chlorophyll fluorescence and calculating the absorbed light energy at PSII, comparing water sprayed and nanoparticle sprayed leaves.
Reviewer 2 Report
Comments and Suggestions for Authors
The study investigates the impact of pegylated zinc-doped ferrite nanoparticles on the photosynthetic efficiency of tomato plants, particularly focusing on their effect on photosystem II (PSII) function. Specifically, tomato plants were sprayed with zinc-doped ferrite nanoparticles at two concentrations (15 mg L⁻¹ and 30 mg L⁻¹), and the impacts on photosystem II (PSII) were evaluated at different time intervals of 30, 60, and 180 minutes using chlorophyll a fluorescence imaging. Key parameters analyzed included the maximum efficiency of PSII photochemistry (Fv/Fm), efficiency of the oxygen-evolving complex (OEC), the distribution of absorbed light energy, non-photochemical quenching (NPQ), electron transport efficiency, PSII excitation pressure, and excess excitation energy, and biphasic hormetic response. While the characterization of the zinc-doped ferrite nanoparticles and the influence of nanoparticles on the efficiency of PSII in response to foliar application are well-documented, several flaws in the study prevent me from recommending it for acceptance.
1. The significance of the findings is not sufficiently compelling for acceptance. The motivation of the study is to improve the photosynthetic efficiency using metal oxide nanoparticles. Yet the results show that ZnFer NPs do not enhance PSII function, particularly after prolonged exposure (greater than 120 minutes). The observed biphasic hormetic response, characterized by a short-term inhibitory effect at low doses and a stimulatory effect at high doses, does not translate into a net improvement in photosynthetic efficiency. Instead, the long-term effect is inhibitory, contradicting the goal of using ZnFer NPs as photosynthetic biostimulants. The logic of the paper would make sense had the ZnFer NPs made a positive impact on the photosynthetic efficiency.
2. The title “Mechanistic Insights on Responses of Tomato Photosystem II Photochemistry to Pegylated Zinc-doped Ferrite Nanoparticles” suggests a focus on understanding the mechanisms driving the observed changes in PSII activity. Despite the promising nature of the title, the research does not sufficiently address the molecular pathways or mechanisms through which zinc-doped ferrite nanoparticles might affect photosynthesis, which would be critical for understanding the potential of these nanoparticles in enhancing plant productivity.
3. While the authors presented the synthesis of a new type of metal oxide nanoparticles, the synthesis procedure or the sample quality does not show enough novelty that suits the scope of the Nanomaterials Journal. The paper will be more suited for plant science journals.
4. The paper can benefit from a comparison between the photosynthetic influence of ZnFer nanoparticles and pristine ferrite nanoparticles. Most importantly, the authors should put in effort to search for other nanoparticle compositions or structures that can truly improve the photosynthetic efficiency. At this stage, the presented results are not sufficient to constitute a meaningful message to the general audience.
In summary, while the study contributes to understanding the effects of ZnFer NPs on photosynthesis, the lack of significant enhancement in photosynthetic efficiency and the absence of mechanistic insights suggest that the findings do not meet the expectations set by the title and abstract. Further research is needed to explore alternative nanoparticle formulations or experimental conditions that could potentially lead to improved photosynthetic efficiency and offer more meaningful mechanistic insights.
Author Response
- The significance of the findings is not sufficiently compelling for acceptance. The motivation of the study is to improve the photosynthetic efficiency using metal oxide nanoparticles. Yet the results show that ZnFer NPs do not enhance PSII function, particularly after prolonged exposure (greater than 120 minutes). The observed biphasic hormetic response, characterized by a short-term inhibitory effect at low doses and a stimulatory effect at high doses, does not translate into a net improvement in photosynthetic efficiency. Instead, the long-term effect is inhibitory, contradicting the goal of using ZnFer NPs as photosynthetic biostimulants. The logic of the paper would make sense had the ZnFer NPs made a positive impact on the photosynthetic efficiency.
Answer: Thank you for your critical evaluation. The study's primary motivation was indeed to explore the potential of ZnFer nanoparticles (NPs) to improve photosynthetic efficiency, specifically through their interaction with PSII function. However, the results indicate a nuanced and complex response that challenges the initial hypothesis. In the revised text several parts have been added in order to present better and more clearly the aim of the study as well as our findings. In so:
At Introduction Part
“Specifically, zinc ferrites (ZnFe₂O₄) have semiconducting properties and can be considered photoreactive materials, including photocatalysis and photochemical reactions. Zinc ferrites NPs have been explored for their photocatalytic activity under UV or visible light [15,16]. In photocatalysis, those NPs can absorb light and generate electron-hole pairs that are capable of driving photochemical reactions, such as the degradation of organic pollutants or the production of hydrogen from water splitting. Meanwhile, the surface of ZnFe₂O₄ can be activated under light, which leads to the formation of reactive oxygen species (ROS), such as hydroxyl radicals (·OH), that can interact with organic contaminants, making them useful in environmental remediation and wastewater treatment.
Herein, in an attempt, to combine two bio-metals with proved crop enhancing and protecting properties, zinc ferrite nanoparticles have been solvothermally prepared in the presence of polyethylene glycol 8000 (PEG 8000) and evaluated on tomato photosystem II photochemistry. Zinc ferrite has a moderate band gap, typically around 1.9–2.2 eV, which makes it responsive to both UV and visible light. This property allows it to participate in photoreactive processes, especially when used in photocatalytic systems. Meanwhile PEG can influence photochemistry through its effects as a stabilizer, or structural component in various systems. It can modulate reaction rates, enhance stability, protect sensitive molecules, and alter the properties of photoreactive materials [77]. Its impact on photochemistry is highly dependent on the specific context of the system and the nature of the photochemical processes involved [78]. We applied a synthetic procedure that allowed the isolation of coated/pegylated ferrites with a zinc doping different from the typical numerical formula (ZnFe2O4), to test further the optical, photocatalytic and magnetic properties of the nanomaterial.”
At Discussion Part
The zinc doping in the ferrite system is expected to enhance the photocatalytic potential of the nanoparticles primarily in terms of absorption of UV light and less effectively concerning utilizing visible light for photochemical or photocatalytic processes [82]. Indeed, the band gap of ZnFer NPs was found slightly larger than the bulk ZnFe2O4 attributed either to nanosize effect or to zinc deficiency. However, ZnFer NPs studied here present a hybrid ensemble, since PEG forms a robust shell on the surface of NPs (70% w/w). PEGylation from the one side allows the better transportation and absorption of the NPs through the plants’ membranes due to the stealth characteristics, but from the other side acts as hindrance in their photocatalytic activity.”
At Conclusions Part
“These findings meet previous results on different types of ferrites(ZnFe2O4 also included) where, in longer growth cycles, seems to be less of an effect from nanoparticles exposure on tomato plant physiological parameters [4]. Thus, both beneficial and detrimental effects in nutritional quality of tomato fruit were induced [4]. This highlights the complex relationship of the type of nanomaterial, concentration, plant species, and growth cycle duration that have to be taking into account in order to use further these nanomaterials for agrochemical purposes. However, the results contribute valuable insights into understanding the biphasic response and the thresholds at which NPs transition from being stimulatory to inhibitory provides critical knowledge for the design of future nanomaterials with improved functionality and reduced adverse effects.”
2. The title “Mechanistic Insights on Responses of Tomato Photosystem II Photochemistry to Pegylated Zinc-doped Ferrite Nanoparticles” suggests a focus on understanding the mechanisms driving the observed changes in PSII activity. Despite the promising nature of the title, the research does not sufficiently address the molecular pathways or mechanisms through which zinc-doped ferrite nanoparticles might affect photosynthesis, which would be critical for understanding the potential of these nanoparticles in enhancing plant productivity.
Answer: We thank the Reviewer for the comment the title has been changed in the revised text to “Response of Tomato Photosystem II Photochemistry to Pegylated Zinc-doped Ferrite Nanoparticles”
3. While the authors presented the synthesis of a new type of metal oxide nanoparticles, the synthesis procedure or the sample quality does not show enough novelty that suits the scope of the Nanomaterials Journal. The paper will be more suited for plant science journals.
Answer: It has to be emphasized that this new type of metal oxide NPs is for the first time reported in such a photosynthetic efficacy study. The synthesis of the ZnFer nanoparticles aimed to balance simplicity, scalability, and functionality for potential applications in plant sciences. While the method itself may not represent a groundbreaking advancement in nanoparticle synthesis, its relevance lies in its suitability for biological applications in plant science. Specifically, the choice of synthesis parameters and characterization were tailored to ensure biocompatibility. The study’s focus on the interaction between ZnFer NPs and PSII function introduces a unique perspective, combining nanomaterials science with plant biology. The interdisciplinary approach of applying these materials to address challenges in photosynthetic efficiency is a valuable contribution that aligns with the journal’s emphasis on nanomaterials with impactful applications.
4. The paper can benefit from a comparison between the photosynthetic influence of ZnFer nanoparticles and pristine ferrite nanoparticles. Most importantly, the authors should put in effort to search for other nanoparticle compositions or structures that can truly improve the photosynthetic efficiency. At this stage, the presented results are not sufficient to constitute a meaningful message to the general audience.
Answer: The consideration of the Reviewer is well accepted, and the revised text has been improved and given to the best of our ability. Moreover, some new references have been added for comparison reasons:
- Kumari, H.; Sonia, Suman, Ranga, R.; Chahal, S.; Devi, S., Sharma, S.; Kumar, S.; Kumar, P.; Kumar, S.; Kumar, A.; Parmar, A review on photocatalysis used for wastewater treatment: Dye degradation. Water Air Soil Pollut. 2023, 234, 349.
- Nguyen, T. B.; Doong, R. Fabrication of highly visible-light-responsive ZnFe₂O₄/TiO₂ heterostructures for the enhanced photocatalytic degradation of organic dyes. RSC Advances, 2016, 6, 103428–103437.
- Bora, U.; Kumar, K.D.; Kumar, S.; Sharma, P.; Nahar, P. Photochemical activation of polyethylene glycol and its application in PEGylation of protein. Process Biochem. 2011, 46, 1380-1383.
- Sheikhi, S.; Aliannezhadi, M.; Tehrani, F.S. The effect of PEGylation on optical and structural properties of ZnO nanostructures for photocatalyst and photodynamic applications. Today Commun. 2023, 34, 105103.
In summary, while the study contributes to understanding the effects of ZnFer NPs on photosynthesis, the lack of significant enhancement in photosynthetic efficiency and the absence of mechanistic insights suggest that the findings do not meet the expectations set by the title and abstract. Further research is needed to explore alternative nanoparticle formulations or experimental conditions that could potentially lead to improved photosynthetic efficiency and offer more meaningful mechanistic insights.
Answer: Our reply that has been added at the Conclusions is given: “These findings meet previous results on different types of ferrites(ZnFe2O4 also included) where, in longer growth cycles, seems to be less of an effect from nanoparticles exposure on tomato plant physiological parameters [4]. Thus, both beneficial and detrimental effects in nutritional quality of tomato fruit were induced [4]. This highlights the complex relationship of the type of nanomaterial, concentration, plant species, and growth cycle duration that have to be taking into account in order to use further these nanomaterials for agrochemical purposes. However, the results contribute valuable insights into understanding the biphasic response and the thresholds at which NPs transition from being stimulatory to inhibitory provides critical knowledge for the design of future nanomaterials with improved functionality and reduced adverse effects.”
Round 2
Reviewer 2 Report
Comments and Suggestions for Authors
I appreciate the authors' efforts in revising the manuscript. While the authors have made efforts to improve the clarity of their study and address previous concerns, significant issues remain that must be resolved before the manuscript can be considered for publication. Below are my detailed comments and recommendations:
1. Significance of Findings and Motivation Misalignment
The study originally aimed to explore the enhancement of photosynthetic efficiency using ZnFer nanoparticles (NPs). However, the results indicate that ZnFer NPs do not improve PSII function, particularly after prolonged exposure, and the observed biphasic hormetic response does not align with the proposed goal of using these nanoparticles as photosynthetic biostimulants. While the authors have expanded the discussion on the complexity of ZnFer NP interactions with PSII, this does not change the fundamental issue.
2. Lack of Comparative Analysis with Pristine Ferrite Nanoparticles
The study would benefit greatly from a comparison between ZnFer NPs and pristine ferrite NPs to clarify whether the observed effects are due to zinc doping or simply inherent properties of ferrite nanoparticles. As the authors mentioned in the manuscript: ‘the surface of ZnFe₂O₄ can be activated under light, which leads to the formation of reactive oxygen species’, and in the response ‘PEG forms a robust shell on the surface of NPs’. Have any studies investigated the photosynthetic influence of ZnFe₂O₄ nanoparticles with any coatings? If not, the authors should at least show the comparison between ZnFe2O4 nanoparticles with PEG and their synthesized ZnFer NPs with PEG to get more in-depth information on how photosynthesis is regulated after the application of the biostimulants.
Given these outstanding concerns, addressing the issues outlined above is necessary to strengthen the manuscript and clarify its contributions to the scientific community.
Author Response
Comment 1: Significance of Findings and Motivation Misalignment
The study originally aimed to explore the enhancement of photosynthetic efficiency using ZnFer nanoparticles (NPs). However, the results indicate that ZnFer NPs do not improve PSII function, particularly after prolonged exposure, and the observed biphasic hormetic response does not align with the proposed goal of using these nanoparticles as photosynthetic biostimulants. While the authors have expanded the discussion on the complexity of ZnFer NP interactions with PSII, this does not change the fundamental issue.
Answer: Our study originally aimed to synthesize via a solvothermal procedure pegylated zinc-doped ferrite (ZnFer) NPs and characterize their physicochemical properties. Subsequently, their impact on tomato photosynthetic efficiency was evaluated by using chlorophyll a fluorescence imaging analysis. From our results we concluded that the synthesized ZnFer NPs, despite their ability to induce hormesis of PSII photochemistry, have a negative impact on photosynthetic function. This was highlighted in the Abstract. In our hypothesis (Lines 150-152) we added: “Since previous studies suggested the safe use of ZnFe2O4 NPs in nano-agricultural applications [9], we evaluated the consequences of our synthesized zinc ferrite NPs on tomato photosynthetic function, 30 min, 60 min, and 180 min after the foliar spray.”
Comment 2: Lack of Comparative Analysis with Pristine Ferrite Nanoparticles
The study would benefit greatly from a comparison between ZnFer NPs and pristine ferrite NPs to clarify whether the observed effects are due to zinc doping or simply inherent properties of ferrite nanoparticles. As the authors mentioned in the manuscript: ‘the surface of ZnFe₂O₄ can be activated under light, which leads to the formation of reactive oxygen species’, and in the response ‘PEG forms a robust shell on the surface of NPs’. Have any studies investigated the photosynthetic influence of ZnFe₂O₄ nanoparticles with any coatings? If not, the authors should at least show the comparison between ZnFe2O4 nanoparticles with PEG and their synthesized ZnFer NPs with PEG to get more in-depth information on how photosynthesis is regulated after the application of the biostimulants.
Answer: According to the best of our knowledge, the use of zinc ferrite (ZnFe₂O₄) nanoparticles either PEGylated or not, has never been reported in other studies at least as biostimulants targeting Photosystem II in tomato leaves. However, related research has explored the effects of ZnFe₂O₄ nanoparticles on other plant species in root and shoot studies. For instance, the reference numbered [9] in our manuscript investigates the impact of naked ZnFe₂O₄ nanoparticles on pea (Pisum sativum) plants and suggests the potential of ZnFe₂O₄ nanoparticles in promoting plant growth. Moreover, the use of PEG has been evaluated in other fields of studies; for example PEG was utilized to functionalize the surface of zinc ferrite nanoparticles synthesized by the hydrothermal method, aiming to improve their biocompatibility and stability for MRI applications (ACS Biomater. Sci. Eng. 2023, 9, 7, 4138–4148). The nanoparticles of this study, were of similar size (14 nm) and quasi-spherical morphology, but with significant lower magnetization value (50 emu/g). While this study focus on medical imaging, the findings imply that magnetization could also play a role in how such nanoparticles interact with plant cellular processes, potentially affecting photosynthetic efficiency. In detail, we added in the discussion part of the revised manuscript:
“Additionally, ZnFer nanoparticles showed enhanced magnetization value, 104 emu/g, compared to PEGylated zinc ferrite nanoparticles (Ms = 50 emu/g) reported in another study for MRI applications [158]. Their strong magnetic properties may influence the generation of ROS, which can either act as signaling molecules or cause oxidative stress. Excess ROS can damage PSII by oxidizing proteins and pigments involved in light harvesting, leading to photoinhibition [35].”
And in the References:
- Dabagh S.; Haris, S.A.; Ertas, Y.N. Engineered Polyethylene Glycol-Coated Zinc Ferrite Nanoparticles as a Novel Magnetic Resonance Imaging Contrast Agent, ACS Biomater. Sci. Eng. 2023, 9, 7, 4138–4148.
Comment 3: Given these outstanding concerns, addressing the issues outlined above is necessary to strengthen the manuscript and clarify its contributions to the scientific community.
Answer: Thank you for your constructive comments which contributed to the improvement of our work.